# Development of a Scalable Process of Film-Coated bi-Layer Tablet Containing Sustained-Release Metoprolol Succinate and Immediate-Release Amlodipine Besylate

**DOI:** 10.3390/pharmaceutics13111797

**Published:** 2021-10-27

**Authors:** Nguyen Thi Linh Tuyen, Le Quan Nghiem, Nguyen Duc Tuan, Phuoc Huu Le

**Affiliations:** 1Faculty of Pharmacy, Can Tho University of Medicine and Pharmacy, 179 Nguyen Van Cu, Can Tho City 94000, Vietnam; 2Faculty of Pharmacy, University of Medicine and Pharmacy at Ho Chi Minh City, 41 Dinh Tien Hoang Street, District 1, Ho Chi Minh City 760000, Vietnam; lequannghiem55@gmail.com (L.Q.N.); ductuan@ump.edu.vn (N.D.T.); 3Department of Physics and Biophysics, Faculty of Basic Sciences, Can Tho University of Medicine and Pharmacy, 179 Nguyen Van Cu, Can Tho City 94000, Vietnam; lhuuphuoc@ctump.edu.vn

**Keywords:** bi-layer tablet, metoprolol succinate, amlodipine besylate, processing conditions, scalable process

## Abstract

The development of new drugs that combine active ingredients for the treatment hypertension is critically essential owing to its offering advantages for both patients and manufacturers. In this study, for the first time, detailed development of a scalable process of film-coated bi-layer tablets containing sustained-release metoprolol succinate and immediate-release amlodipine besylate in a batch size of 10,000 tablets is reported. The processing parameters of the manufacturing process during dry mixing-, drying-, dry mixing- completion stages were systematically investigated, and the evaluation of the film-coated bi-layer tablet properties was well established. The optimal preparation conditions for metoprolol succinate layer were 6 min- dry mixing with a high-speed mixer (120 rpm and 1400 rpm), 30-min drying with a fluid bed dryer, and 5 min- mixing completion at 25 rpm. For the preparation of amlodipine besylate layer, the optimal dry-mixing time using a cube mixer (25 rpm) was found to be 5 min. The average weight of metoprolol succinate layers and bi-layer tablets were controlled at 240–260 mg and 384–416 mg, respectively. Shewhart R chart and $\bar{X}$ charts of all three sampling lots were satisfactory, confirming that the present scalable process was stable and successful. This study confirms that the manufacturing process is reproducible, robust; and it yields a consistent product that meets specifications.

## 1. Introduction

High blood pressure (hypertension) is a common condition in which the long-term force of the blood against your artery walls is high enough that it may eventually cause health problems. An estimated 1.39 billion (1.34–1.44 billion) adults aged ≥ 20 years worldwide had hypertension, 694 million (659–730 million) men and 694 million (660–727 million) women, in 2010 [1]. Almost three times as many individuals with hypertension lived in low- and middle-income countries (1.04 billion [0.99–1.09 billion]) than in high-income countries (349 million [337–361 million]) [1]. Hypertension can promote coronary artery disease, heart failure, cerebrovascular disease, chronic kidney disease, and reduced quality of life. More severely, over 19% of all deaths in 2015 were linked to elevated systolic blood pressure (>115 mm Hg) [2], and hypertension also can cause of severe neurological sequelae such as hemiplegia, and lethargy with plant life [2]. For the treatment of hypertension, monotherapy is the primary standard treatment for controlling blood pressure in most patients. However, recent studies show that a combination of drugs is an effective method to well-control blood pressure [3,4]. According to the European Society of Cardiology and European Society of Hypertension (ESC/ESH) in 2018, most patients need at least two drugs to achieve the treatment goals, and the treatment can be monotherapy or a combination of two drugs in low doses [4]. The combined drugs have been found to increase the effectiveness of hypotension, improve patient tolerance, reduce side effects and cardiovascular events [5]. Particularly, a combination of β_1_-adrenergic blockers and calcium channel blockers drugs are the primary choice for hypertension treatment with heart failure (e.g., coronary artery disease), and for patients who do not achieve the effective result with monotherapy using β_1_-adrenergic blockers or calcium channel blockers.

Metoprolol succinate is a selective β_1_-adrenergic antagonist, while amlodipine besylate is a calcium channel inhibitor of the dihydropyridine group. The combined therapy of the two ingredients can be expected to effectively treat hypertension with coronary artery disease [6]. Since metoprolol succinate is strongly metabolized by liver enzymes and has a short half-life of 3–4 h and low bioavailability of 40%, it causes inconvenience in usage when the drug has to be used several times a day [7]. In contrast, amlodipine besylate is a drug with a long half-life of 30–40 h [7]. It is challenging to successfully combine the two ingredients due to such far differences in half-life and release mechanism to develop a new drug product, which has sustained-release metoprolol succinate and immediate-release amlodipine besylate to stabilize blood pressure in 24 h and reduce unwanted side effects.

Amlodipine besylate is an active ingredient that is easily hygroscopic and unstable (i.e., modified easily under direct light exposure). Meanwhile, metoprolol succinate is a good water-soluble active ingredient, and consequently, it is difficult to select a suitable polymer for controlling its sustained release. In some countries around the world, the combined tablets containing metoprolol succinate and amlodipine besylate have been developed and used with brand name drugs such as Selomax^TM^, Sitelol, but the detailed technological procedures to produce these drugs have not been reported. In Vietnam, the combined drug has not been developed, and patients have to use several individual ingredient tablets or use the import combined drugs for their treatment needs that can increase treatment costs and inconvenience.

Among various tablet manufacturing techniques, direct compression has been realized as the most practical and cost-effective technique. For this process, tablets are obtained from the mixture of active ingredients and diluents, followed by the addition of lubricants, disintegrants, and finally, by compression of the final mixture [8]. In addition, blending is an important and common manufacturing process for preparing a solid dosage form (i.e., tablets and capsules) of pharmaceutical drugs. Blending is performed primarily in a rotating device (e.g., cubic blender, fluidized bed dryer) [9]. To ensure the quality of solid dosage forms, the mixture requires achieving a degree of homogeneity during blending that involves key processing conditions of drying and mixing time. For developing film-coated bi-layer tablets, the manufacturing process becomes even more challenging because of the required optimization of larger various processing equipment conditions.

In our previous study, film-coated bilayer tablets containing sustained-release metoprolol succinate and immediate-release amlodipine besylate were prepared on a laboratory scale of 400 tablets [10]. The effects of polymers and fillers in the formulation and compression force on the percentage of drug released from the film-coated bi-layer tablets [10]. However, the successful manufacturing process on the laboratory scale (e.g., 400 tablets/lot) cannot directly apply to the process of a larger scale (e.g., 10,000 tablets/lot) because of the use of different specialized equipment for different bath sizes. The quality of granules might change associated with the process. Indeed, moisture content, friability, and granule size distribution can significantly affect the final tablet properties such as tablet hardness, the dissolution rate of the active ingredient, etc. [11]. In this study, for the first time, we report the formulation and detailed production process of film-coated bi-layer tablets containing sustained-release metoprolol succinate and immediate-release amlodipine besylate at a scale of 10,000 tablets. Noticeably, the Shewhart chart is used to control the tablet weight of three batches, which is very important for controlling the compression process of the bi-layer tablets. This study provides the detailed key optimum conditions and gains insight into the inter-relationship between processing conditions and the quality of the drug products. Moreover, the introduced process could allow reducing the research and studying time toward achieving high stability of the mixture. Furthermore, the results of this study contribute to the development of new domestically produced drugs to meet the high and diversified treatment demands and replacement of imported drugs.

## 2. Materials and Methods

### 2.1. Materials

The active pharmaceutical ingredients were metoprolol succinate (Polpharma S.A., Duchnice, Poland), amlodipine besylate (Cadila Healthcare Limited, Ahmedabad, India). Hydroxypropyl methylcellulose (Methocel K100M), Pregelatinized Maize Starch (Starch 1500) were kindly donated by Colorcon, China. Aerosil (Evonik Industries AG, Essen, Germany), Glucidex (Roquette Pharma, Gurnee, IL, USA), and Tablettose (Meggle Pharma, Wasserburg am Inn, Germany) were kindly donated by Chemical Company Limited (Dang Hung, Ho Chi Minh City, Vietnam). Comprecel M101LD (Mingtai Chemical Co., Ltd., Taoyuan, Taiwan), Di-tab (Reephos Chemical Co., Ltd., Lianyungang, China), Polyvinylpyrrolidone (PVP K30, ISP, Lewes, DE, USA), Xanthan gum (Jungbunzlauer Suisse AG, Switzerland), and sodium croscarmellose (Mingtai Chemical Co., Ltd., Taoyuan, Taiwan) were purchased from DHG Pharmaceutical Joint–Stock company (Can Tho City, Vietnam). All materials used in this study complied with current United States Pharmacopeia–National Formulary (USP-NF) compendial specifications. 

Metoprolol succinate (99.94%, purity) (Batch No. 98418-47-4) as the internal standard was obtained from Polpharma S.A., Duchnice, Poland. Amlodipine besylate (100.43%, purity) (Batch No. QT.145090516) as the internal standard was purchased from the Institute of Drug quality control in Ho Chi Minh City, Vietnam. Mobile phase components for high-performance liquid chromatography (HPLC) were of analytical grade. All other chemicals meet the analytical standards and were purchased commercially.

### 2.2. Composition of Tablet in the Batch Size of 10,000 Tablets

The formulation of film-coated bi-layer tablets containing sustained-release metoprolol succinate and immediate-release amlodipine besylate on a laboratory scale was first reported in Ref. [6] and in our earlier study [10]. Herein, for the first time, we report the detailed formula and production process of the drug in a batch size of 10,000 units. The formula for each tablet is summarized in Table 1.

### 2.3. Preparation Processes of Sustained-Release Metoprolol Succinate Layer and Immediate-Release Amlodipine Besylate Layer

The sustained-release metoprolol succinate layer was prepared by the wet granulation method. Metoprolol succinate and other excipients like starch 1500 and di-tab (a diluent), HPMC K100M and xanthan gum (a matrix former) were accurately weighed and passed through a sieve (#50-mesh) to ensure a disaggregated state prior to mixing. Alcohol (96%) was used as a solubilizing agent, and PVP K30 was used as the binding agent. To prepare the PVP solution, PVP K30 was added and gently dissolved in an alcohol (96%) at ambient temperature (20–25 °C) until the solution goes clear. Metoprolol succinate, HPMC K100M, xanthan gum, starch 1500, and di-tab were mixed in a high-speed mixer, and the PVP solution was added slowly to get cohesive mass. The mixing and granulation was performed in a high-speed mixer NT-5 (TienTuan, Ho Chi Minh City, Vietnam) with a capacity of 5 kg. The impeller and chopper speeds were set at 120 rpm and 1400 rpm, respectively, while the mixing time was investigated. The wet granules were dried using a fluidized bed dryer FBDG 2-5 (TienTuan, Ho Chi Minh City, Vietnam) at conditions of inlet air temperature 60 °C and hot-air flow rate of 150 m^3^/h until its moisture content reached 3–5%. Finally, the granules were transferred to a stainless steel cube mixer CB-5 (TienTuan, Ho Chi Minh City, Vietnam) with a capacity of 5 kg and a fill weight of approximately 50% of the mixer volume. Aerosil and magnesium stearate were added into the cube mixer rotating at 25 rpm to obtain a homogeneous mixture.

The immediate-release amlodipine besylate layer was prepared by direct compression method. Amlodipine besylate and other excipients of tablettose (a diluent), sodium croscarmellose (a superdisintegrant), glucidex (a binder), aerosil and magnesium stearate (lubricants) were accurately weighed and passed through a sieve (#50-mesh) to ensure a disaggregated state prior to mixing. All formulation materials, except aerosil and magnesium stearate, were mixed in a stainless steel cube mixer CB-5 (TienTuan, Ho Chi Minh City, Vietnam) with a capacity of 5 kg and a fill weight of approximately 30% of the mixer volume. The cube mixer was rotated at a speed of 25 rpm for several investigated mixing times. Aerosil and magnesium stearate were added to the above mixture and mixed thoroughly to obtain a homogeneous mixture.

### 2.4. Optimization of the Mixing Time and the Drying Time

Three batches of 10,000 tablets/batch were prepared and investigated to optimize the processing parameters, including mixing time and drying time. Detailed the processing conditions of mixing and drying phases were presented in Table 2.

We evaluated the homogeneity of the mixture via a coefficient of variation (CV% ≤ 2%) by quantitative analysis of the active substances in the samples at different time points during the dry mixing and dry mixing completion stages. The mixing and drying times resulted in the smallest CV% is preferred as the optimal time conditions.

Herein, the method for simultaneous quantitative analysis of metoprolol succinate and amlodipine besylate was developed by N.T.L Tuyen et al. [12,13] using HPLC (Agilent Technologies, Wood Dale, IL, USA) with the following chromatographic conditions: column Xterra^®^ RP18 (250 × 4.6 mm; 5 µm), PDA detector, UV wavelength of 230 nm. The mobile phase was a mixture of acetonitrile and aqueous phosphoric acid (pH = 4) with a ratio of 32:68 (*v/v*), then it was filtered through a 0.45 μm- membrane filter and degassed. The analysis was performed at room temperature, a flow rate of 0.6 mL/min, and an injection volume of 20 µL [12,13].

### 2.5. Evaluation of Physical Properties of Granules

Prior to compression, according to USP 43–NF 38 [14], the evaluated parameters of granules include moisture content, bulk density, tapped density, compressibility index, Hausner’s ratio, and angle of repose.

Moisture analyzer MA35 (Sartorius AG, Göttingen, Germany) equipped with a halogen lamp was a device that determined the moisture content [15]. The bulk and tapped density of the granules were determined using a PT-TD300 instrument (Pharma-Test, Hainburg, Germany). Bulk density (p_b_) was the weight of granules divided by its volume. Tapped density (p_t_) was the weight of granules divided by its tapped volume. The compressibility index (CI) and Hausner’s ratio (H) were calculated using the following Equations (1) and (2):(1)$$\mathrm{CI}=\frac{p_{t}-{\;p}_{b}}{p_{t}}\times 100$$
(2)$$H=\frac{p_{t}}{p_{b}}$$

The angle of repose was estimated by the fixed funnel method. The weighed final mixture was taken in the funnel. The bottom of the funnel was opened and the granule was allowed to flow freely to form a smooth conical heap. The radius of the heap (r) and the height of the heap (h) were measured. The value of the angle of repose (θ) was calculated using the following Equation (3):(3)$$\tan\theta =\frac{h}{r}$$

### 2.6. Process of bi-Layer Tablet Compression and Film Coating

#### 2.6.1. Process of bi-Layer Tablet Compression 

We used the double rotary tablet compression machine 2-DV-5 (Royal Pharma, Boisar, India) and oval-shaped punches (9 × 12 mm) to compress the metoprolol succinate layer with an average tablet weight of 250 mg ± 4% (240–260 mg) under a pre-compression force of 3 kN. We employed the Shewhart chart to control the average tablet during the compression process [16,17]. We also adjust the average weight of tablets to 400 ± 4% mg (384–416 mg) via the following assessment rules:

Rule 1 (point outside the 3σ control limit): 1 point outside the control limit line. The Shewhart R chart considers only when the upper limit line occurs.

Rule 2 (trend toward one value): 7 consecutive points, 10 on 11 consecutive points, 12 on 14 consecutive points, or 16 on 20 consecutive points on the same size of the center line. The Shewhart R chart considers only when an upward trend occurs.

Rule 3 (tendency to increase or decrease): 6 consecutive points are steadily increasing or decreasing. The Shewhart R chart considers only when the upper control line occurs.

Rule 4 (trend increase and decrease in cycles): 14 consecutive points alternating up and down.

Rule 5 (tendency to close the 3σ control limit): 2 out of 3 consecutive points are more than 2σ from the center line in the same direction. The Shewhart R chart considers only when the upper limit line occurs.

Rule 6 (tends to be outside the 1σ boundary): 4 out of 5 consecutive points are more than 1σ from the center line in the same direction [16,17].

#### 2.6.2. Process of bi-Layer Tablet Film Coating

The formed bi-layer tablet was coated by a film prepared from Opadry II 85F19250 Clear suspension 8% (*w*/*w*) in an alcohol–water solution. The components of the film coating were presented in Table 1. The film coating material was prepared by slowly adding Opadry II into a beaker containing an alcohol–water solution under magnetic stirring for 15 min, and then the product was sifted through a 0.4 mm sieve to obtain the film coating suspension. Notably, the suspension was well stirred during the coating process. The film coating process was performed using perforated sugar-coating pan equipment under conditions of pan speed of 10–15 rpm, inlet air temperature of 60–70 °C, core temperature of 40–45 °C, spray rate of 1–2 mL/min, and atomization air pressure of 1.5 Pa. The distance between the gun and the surface of the coated tablet was 17 cm, and spray width covered the entire width of the bi-layer tablet. Afterward, the thermal curing process was performed at 50 °C for 30 min.

### 2.7. Pharmaceutical Quality Evaluation of Film Coated bi-Layer Tablet


**Appearance**


The general appearance of a bilayer tablet was identified by visually observing its shape, colour, and surface texture.


**Weight variation**


To study the weight variation of film-coated bi-layer tablet, a set of 20 tablets from each formulation were weighed using a digital precision balance (Mettler Toledo PL303, Greifensee, Switzerland, readability of 0.001 g), and the test was performed according to the official method, considering a weight variation limit of ± 5% [14,18,19].


**Content uniformity**


A set of 10 tablets from each formulation was randomly collected to determine the content of amlodipine besylate in each film-coated bi-layer tablet using the HPLC method. Detailed key parameters of the HPLC analysis method and its validation are reported in Appendix A. We found that amlodipine besylate content in every tablet was in the range of 85–115% of the averaged content. The product fails to meet the requirement if there is more than one tablet having amlodipine besylate out of the above range, or if there is one tablet containing amlodipine besylate out of the 75–125% range. If there is an amlodipine-besylate-containing tablet out of the 85–115% range, the other 20 tablets will be tested. Inversely, the product achieves the requirement if there is no more than one tablet having amlodipine besylate out of the 85–115% range among the 30 tested tablets, and none of those having amlodipine besylate out of the range of 75–125% of the averaged content [14].


**Hardness and friability**


The hardness of randomly selected bi-layer tablets (*n* = 20) was determined using the USP method with a hardness tester (Erweka TBH 125, Germany). The weight of the bi-layer tablets (*n* = 20) was noted initially as W1, and they were rotated in a friability tester (Erweka TAR 120, Heusenstamm, Germany) at speed of 25 rpm for 4 min. Then, the tablets were reweighed and noted as W2. The weight difference in percentage [(W1 − W2)·100/W1] was noted and expressed as a percentage [20].


**In vitro dissolution study**


The drug release from different batches of the prepared tablets was carried out using the USP dissolution apparatus type II (paddle method). The used dissolution medium was 500 mL of HCl 0.01 N and a paddle speed of 75 rpm for the first 30 min for immediate-release amlodipine besylate layer and then 500 mL of phosphate buffer (pH = 6.8) and a paddle speed of 50 rpm were used up to 20 h for sustained-release metoprolol succinate layer. During the test, the medium temperature was maintained at 37 ± 0.5 °C. Sample aliquots (10 mL) were withdrawn at several proper time intervals (e.g., 30 min for immediate-release amlodipine besylate layer; 1 h, 4 h, 8 h and 20 h for sustained-release metoprolol succinate layer), and then filtered through a 0.45 μm membrane filter. The drug contents were analyzed using HPLC (Agilent Technologies, USA) at a UV detection at a wavelength of 230 nm [12]. The cumulative percentage of release drug was calculated and the results were presented as the mean value of at least six tablets. It was required that all tablets meet the required limitations of the releasing percentages of amlodipine besylate and metoprolol succinate, as reported in Appendix A.


**Drug content**


Twenty tablets from each batch containing metoprolol succinate and amlodipine besylate were selected randomly, accurately weighed, and grounded to a fine powder. Cumulative drug release was used to determine the content of metoprolol succinate and amlodipine besylate via the HPLC method [12]. The average content percentages of both metoprolol succinate and amlodipine besylate were within the range of 90–110% of the label content.

## 3. Results and Discussion

### 3.1. Determination of the Mixing Time and the Drying Time of Granules

The homogeneity of the dry mixing stage, drying stage, and dry mixing completion stage of batch 1 was evaluated from the results of dispersion analysis of metoprolol succinate, as summarized in Table 3.

The dry mixing stage is a mixing ingredient process in the high-speed mixer at an impeller speed of 120 rpm and chopper seeding speed of 1400 rpm. As shown in Table 3, a mixing time of 6 min ensures well dispersion of the active ingredients in the powder and offers the smallest CV (%) as compared to those of the other mixing times (i.e., 4 or 8 min). For the drying phase of the same batch, granules needed cool air blown for 10 min to evaporate the alcohol, to avoid fire, and explosion factors in the production process, and then it was dried at 50 °C for 30 min to achieve a moisture content of 3–5%. This drying time was really short as compared with the drying time using a drying oven (>6 h). For the dry mixing completion stage, an optimal time was 5 min, where metoprolol succinate content had the smallest CV value of 0.85%. Furthermore, the content percentages of metoprolol succinate in the three processing stages of three batches are presented in Table 4.

The equipment and mixing time of powders can greatly affect the mixture homogeneity and the homogeneity of tablets. Indeed, they can vary the granule properties such as particle size distribution, bulk density, tapped density, compressibility index, Hausner’s ratio, and angle of repose. In addition, the granule moisture content affected the flow property and the degree of particulate bonding. The granules were dried to a moisture content of 3–5% so that the granules are flowable and obtain the required hardness.

Similar to the metoprolol succinate layer, the homogeneity evaluation of the dry mixing stage and dry mixing completion stage of batch 1 was studied by analyzing the dispersion of amlodipine besylate content. The results were presented in Table 5.

The optimal dry mixing time was found to be 5 min because it gave the highest mean amlodipine besylate content of 100.91% and the smallest CV of 0.71%. When upgrading the batch size to 10,000 tablets, the use of a cube mixer for amlodipine besylate in the dry mixing and mixing completion stages allowed shortening the study time and obtaining high homogeneity of the mixture. Detailed preparation and results of the amlodipine besylate layer on batch 1, 2 and 3 are presented in Table 6.

For a scalable production process, a high-speed mixer is a high-efficiency equipment that can mix different powder materials and granulates in one procedure. During the dry mixing stage, the cohesive powder components had to be disagglomerated to obtain a high degree of mixing [21,22]. Nonuniformity may occur in a certain period during the mixing process that needs to be accurately determined. Dry mixing parameters were used by our experience and following the process testing result in ref. [9]. The duration and mixing speed were factors that influenced the material mixing process and the homogenization of the mixing results [9,21]. The impeller speed range was 100–1500 rpm and the chopper speed range was 1000–3000 rpm. A dry mixing time of approximately 6 min was suggested by the machine manufacturer [21,22]. This was true for selecting the mixing time and mixing speed on the high-speed mixer.

In the pharmaceutical industry, a fluidized bed dryer is utilized for drying as a step after wet granulation because a fluidized bed dryer gives a faster drying rate as compared to a drying oven [23]. The granules can achieve high homogeneity in terms of moisture content, bulk density, and tapped density [23,24].

A cube mixer was used for dry-mixing, dry-mixing completion stages of amlodipine besylate powder mixture, and dry-mixing completion of metoprolol succinate granules. The process of powder blending is influenced by diffusional and convective forces [11,21]. Following an increase in blending duration, it is likely that the movement of powder bed through convection increases the distribution of drug particles between the excipients to result in the generation of a random blend. An increase in blending duration also ensures that diffusional blending promotes the movement of particles, thereby enhancing content uniformity [9]. For the two blending techniques, the use of a cube mixer enables the production of homogeneous blends under a processing time of 5 min, meanwhile, manual blending at the lab-scale failed to achieve the compendial requirement of content uniformity under mixing time up to 15 min [21].

For a bilayer manufacturing process, the use of specialized equipment, namely a high-speed mixer in the dry mixing stage of metoprolol succinate mixture, fluidized bed dryer for granule drying, and cube mixer in the dry mixing completion phase of amlodipine besylate mixture and metoprolol succinate granule allowed to shorten the time of the study. They enable achieving a high uniformity mixture and meet the requirements of intermediate product testing criteria. The reported process is highly productive and efficient that allows the production of the bi-layer tablet in a short time and reduced production costs on a large scale.

### 3.2. Evaluation of Physical Properties of Granules

The results of intermediate products of metoprolol succinate and amlodipine besylate in three batches are presented in Table 7. Evaluation parameter of granules such as moisture content, bulk density, tapped density, compressibility index, Hausner’s ratio, and angle of repose.

The bulk density of granules was found to be between 0.461 g/mL and 0.755 g/mL. The values indicate good packing characteristics [25]. The tapped density of granules of three batches was found in the range of 0.499 g/mL to 0.834 g/mL. The compressibility index (CI) of each batch was found to be 7.801–9.484% indicating excellent flow properties of granules (5–15%) which were further confirmed by determining the angle of repose. The angle of repose values of metoprolol succinate and amlodipine besylate granules were in narrow ranges of 23.35°–23.39° and 28.97°–28.99°, respectively, indicating that the granules obtained excellent flow ability. Hausner’s ratio was found to be in the range of 1.084–1.105 for both metoprolol succinate and amlodipine besylate. These values are below 1.25, confirming a good flow property [25,26].

### 3.3. Process of Tablet Compression

The sustained-release metoprolol succinate layer was added to funnel No. 1, while the immediate-release amlodipine besylate layer was added to funnel No. 2. The tablet compression was conducted using double rotary tablet press machine 2-DV-5 (Royal Pharma, India), oval-shaped punch (9 × 12 mm), at a speed of approximately 12 rpm. The metoprolol succinate layer had an average weight of 250 mg/tablet ± 4% (240–260 mg) under a pre-compression force of 3 kN. Then, we adjusted the tablet weight to 400 mg ± 4% (from 384 to 416 mg) by using funnel No. 2, applied a pressing force of 10–12 kN, and maintained friability under 1%. The interval between two sampling times was 30 min.

Compression force is an important parameter for forming bi-layer tablets [10,27]. When the compression force increases, the mechanical strength of the tablet increases, while the porosity and microcapillary system in the bi-layer tablet decreases. Therefore, the bi-layer tablet becomes more difficult to absorb water and prolongs the disintegration time, which results in the decrease in the drug-releasing percentage of the tablet [19,27]. The present bilayer tablet compression strategy was similar to that reported in ref. [28,29], where a compressive force was first applied to the sustained-release layer (i.e., metformin hydrochloride), then an immediate-release layer (i.e., sitagliptin phosphate) was introduced, and finally, the whole tablet was compressed to complete the process [28]. The present compression strategy is attributed to the high hardness values between 10.36 ± 0.68 kp and 10.48 ± 0.57 kp of the bilayer tablets and allows obtaining the desired drug release percentage and in vitro dissolution of the bilayer tablets.

The average weight was evaluated through the Shewhart R/$\bar{X\;}$ chart. The surveyed results of the Shewhart R/$\bar{X}$ chart on the tablet weight of three batches are shown in Figure 1. The Shewhart R and $\bar{X\;}$ chart results for tablet weight of the three batches are also summarized in Table 8 and Table 9.

Bi-layer tablet is suitable for sequential release of two drugs in a combination form of two incompatible substances, in which one layer was immediate-release as initial dose and the second layer was maintenance dose [11,18,25]. Compression force-controlled presses were clearly limited when a quality bi-layer tablet needed to be produced in conjunction with accurate weight control of both layers. Low pre-compression forces were necessary to secure interlayer bonding. The use of a higher compression force in the second layer may rapidly result in separation and hardness problems when compressing bi-layer tablets [11]. Therefore, the produced tablets’ quality fulfilled all product specifications [23].

In the compression stage, we used an assemble of 9 × 12 mm oval punch with a speed of 12 cycles per minute. The average weight of the tablet was controlled during the compression process via the Shewhart R/$\bar{X\;}$ chart. The interval between two sampling times was 30 min. The compression time for each batch ranged from 11 to 12 h. In addition to controlling the average mass of tablets, the average layer of tablets was also controlled. This was a critical issue in process control for formulation bi-layer tablets because if we only control the mass of tablets, we cannot guarantee the accuracy of the target mass of each layer. During the production deployments, the mass of each layer can be simply checked by observing the thickness of the metoprolol succinate layer. It was found that the mass of the metoprolol succinate layer was likely to be high or low when the layer thickness is over 4 mm or under 3.5 mm, respectively. Shewhart R and $\bar{X}$ charts of all three batches were satisfactory, confirming that the bi-layer tablet production process was stable and successful in a scalable process.

### 3.4. Pharmaceutical Quality Evaluation of Film Coated bi-Layer Tablet

The results of the film-coated bi-layer tablet of three batches were found to be within limits (weight variation ±5%, content uniformity ±15%, hardness range 10.36–10.48 kp, friability <1%, and drug content 90–110%), and all the values are reported in Table 10.

All film-coated bi-layer tablet batches were white-colored, oval-shaped, biconvex tablets with a smooth surface, and the two layers could be clearly distinguished based on the color difference (Figure 2). There was no chipping or mottling observed in any of the formulated tablets. The weight variation of the film-coated bi-layer tablet was found to be uniform (410.42 ± 0.85 mg). According to USP, for 400 mg tablets, not more than two tablets differ from the average weight by 5%, and no tablet differs by more than double the relevant percentage [14]. Content uniformity for all the prepared formulation batches was from 4.72 mg to 4.88 mg for amlodipine besilate which were within the limits.

The mean hardness of the prepared tablets was 10.43 kp, while the observed percentage of friability was 0.14, which was indicated as good regarding the strength of the tablet. The study has reported that there is a decrease in the percentage of friability with an increase in tablet hardness [30]. All batches were within the compendial limit of <1%. The harder the tablet, there will be fewer chances of chipping and breakage [11]. The drug content of the film-coated bi-layer tablets was found to be in the range of 93.56–96.97% for both metoprolol succinate and amlodipine besylate.

From Table 11, amlodipine besylate was released from 94.39% to 98.66% at 30 min, while metoprololol succinate was released from 86.21% to 87.25% at the end of 20 h. the percent drug release of metoprololol succinate after the first hour was found between 12.69% and 14.01%. The tablet does not alter the dissolution of metoprolol succinate or amlodipine besylate based on the evaluated results of three batches. The results of this study indicate that film-coated bi-layer tablets containing sustained-release metoprolol succinate and immediate-release amlodipine besylate, on a scale of 10,000 tablets, were successfully produced by our reported process. The drugs should be used as an antihypertensive agent and for sustaining the antihypertensive activity.

## 4. Conclusions

We report the manufacturing process of film-coated bi-layer tablets containing sustained-release metoprolol succinate and immediate-release amlodipine besylate on a scale of 10,000 tablets/lot. Particularly, the optimal processing conditions for metoprolol succinate layer were 6-min- dry mixing with a high-speed mixer at a speed of 1400 cycles/min, 30 min drying stage with a fluid bed dryer, and 5 min mixing completion stage at a speed of 25 rpm. In addition, the amlodipine besylate layer underwent a dry mixing process in a cube mixer for 5 min at a speed of 25 rpm. The average mass metoprolol succinate layer was 250 mg/tablet ± 4% (240–260 mg), and the film-coated bi-layer tablet mass was controlled at 400 mg ± 4% (384–416 mg). The film-coated bi-layer tablet production process at the scale of 10,000 tablets was successful and stable as proven by the satisfactory results of the Shewhart R and $\bar{X}$ charts for all three investigated batches. The products of all three batches at the scale of 10,000 tablets obtained a unified quality, the required dissolution. The present scalable manufacturing process is reproducible and robust to yield consistent product, which meets specifications. The study is greatly valuable to those working on formulations and optimization manufacturing processes of film-coated bi-layer tablets on various manufacturing scales. It also contributes to fundamental knowledge and understanding for further developments of the tablets on a larger scale.

## Figures and Tables

**Figure 1 pharmaceutics-13-01797-f001:**
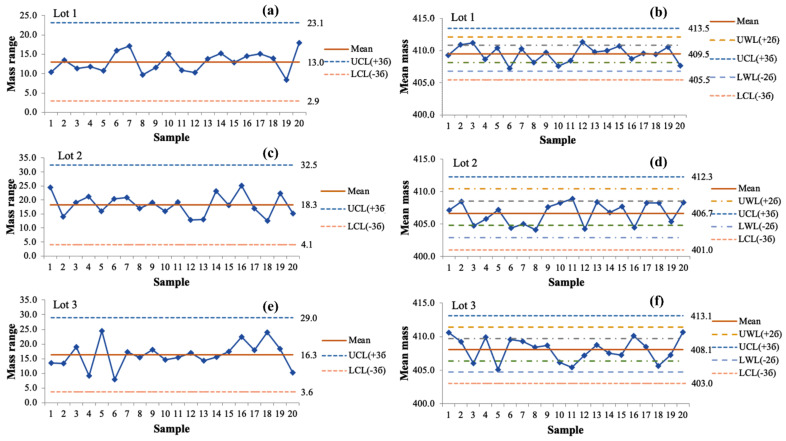
Shewhart mass range (R) and mean mass ($\bar{X}$) diagrams of tablet mass of three batches. (**a**,**b**) Shewhart R/$\bar{X}$ chart of lot 1, (**c**,**d**) Shewhart R/$\bar{X}$ chart of lot 2, (**e**,**f**) Shewhart R/$\bar{X}$ chart of lot 3.

**Figure 2 pharmaceutics-13-01797-f002:**
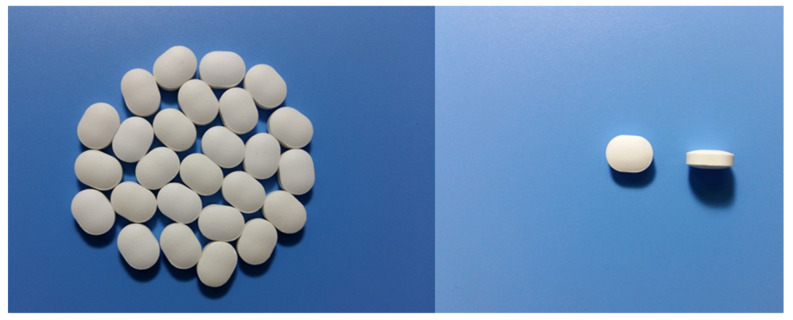
The appearance of the as-prepared film coated bilayer tablets.

**Table 1 pharmaceutics-13-01797-t001:** Formula for film coated bi-layer tablet.

Ingredients	Weight per Tablet (mg)	Weight for 10,000 Tablets (g)
** *Sustained-release* ** ** *layer* **		
Metoprolol succinate *	47.500	475.00
Starch 1500	8.333	83.33
Di-tab	4.167	41.67
HPMC K100M	135.000	1350.00
Xanthan gum	45.000	450.00
PVP K30	5.000	50.00
Aerosil	2.500	25.00
Magnesium stearate	2.500	25.00
96% Alcohol ***		850
Total	250	2500
** *Immediate-release layer* **		
Amlodipine besylate **	6.935	69.35
Tablettose	126.565	1265.65
Sodium croscarmellose	3.000	30.00
Glucidex	10.500	105.00
Aerosil	1.500	15.00
Magnesium stearate	1.500	15.00
Total	150	1500
** *Components of f* ** ** *ilm coating* **		
Opadry II 85F19250 Clear	6.000	60.00
96% Alcohol ***		450
Water ***		240
Total	6	60

* 47.5 mg of metoprolol succinate is equivalent to 50 mg of metoprolol tartrate and 19.5 mg of metoprolol. ** 6.935 mg of amlodipine besylate is equivalent to 5 mg of amlodipine. *** Solvents easily evaporate during preparation and hardly exist in the product.

**Table 2 pharmaceutics-13-01797-t002:** Preparation conditions for metoprolol succinate layer and amlodipine besylate layer in scale of 10,000 tablets/batch.

Processing Stage	Metoprolol Succinate Layer	Amlodipine Besylate Layer
Dry mixing stage	Equipment: high speed mixer (5 kg)	Equipment: cube mixer (5 kg)
	Impeller speed: 120 rpm Chopper speed: 1400 rpm	Round speed: 25 rpm
	Sampling location: 6 different positions	Sampling location: 6 different positions
	Volume of samples taken: 2 g/time/position × 3 times	Volume of samples taken: 2 g/time/position × 3 times
	Time of sampling: 4, 6, 8 min	Time of sampling: 3, 5, 7 min
Drying stage	Equipment: fluid bed dryer (5 kg)	
	Sampling at 3 times: 10, 20, 30 min	
	Volume of samples taken: 2 g/time/position × 3 times	
	Requirements: moisture content of 2–4%	
Dry mixing Completion stage	Equipment: cube mixer (5 kg)	Equipment: cube mixer (5 kg)
	Round speed: 25 rpm	Round speed: 25 rpm
	Sampling location: 6 different positions	Sampling location: 6 different positions
	Volume of samples taken: 2 g/time/position × 3 times	Volume of samples taken: 2 g/time/position × 3 times
	Time of sampling: 3, 5, 7 min	Time of sampling: 3, 5, 7 min

**Table 3 pharmaceutics-13-01797-t003:** Processing time during dry mixing-, drying stages of metoprolol succinate layer at different locations (L) in batch 1.

Stage	Time (min)	Metoprolol Succinate Content (%)							
		L1	L2	L3	L4	L5	L6	Mean	CV (%)
Dry mixing	4	97.06	96.52	100.49	98.23	96.49	99.72	98.09	1.59
	6	99.55	100.04	99.83	99.28	100.78	100.37	99.98	0.50
	8	99.63	103.47	101.30	102.39	99.10	100.68	101.10	1.49
Drying	10	8.66	8.52	8.40	-	-	-	8.53	1.24
	20	5.09	5.18	5.12	-	-	-	5.13	0.73
	30	3.60	3.58	3.60	-	-	-	3.59	0.26
Dry mixing Completion	3	102.69	101.32	99.00	99.03	101.13	99.66	100.47	1.47
	5	99.98	101.49	99.74	99.70	100.70	99.08	100.12	0.85
	7	100.58	101.92	99.58	99.07	101.22	100.85	100.54	1.05

Coefficient of variation (CV).

**Table 4 pharmaceutics-13-01797-t004:** Optimal process of metoprolol succinate layer in three batches.

Stage	Batch	Metoprolol Succinate Content (%)							
		L1	L2	L3	L4	L5	L6	Mean	CV (%)
Dry mixing	1	99.55	100.04	99.83	99.28	100.78	100.37	99.98	0.50
	2	100.17	99.45	98.24	100.20	100.51	98.23	99.47	0.93
	3	99.41	99.98	100.19	101.32	99.02	100.31	100.04	0.73
Drying	1	3.60	3.58	3.60	-	-	-	3.59	0.26
	2	3.68	3.66	3.70	-	-	-	3.68	0.44
	3	3.62	3.62	3.60	-	-	-	3.61	0.26
Dry mixing Completion	1	99.98	101.49	99.74	99.70	100.70	99.08	100.12	0.85
	2	97.89	99.83	100.37	100.33	99.95	100.30	99.78	0.95
	3	101.13	100.85	101.92	101.28	100.28	99.82	100.88	0.74

**Table 5 pharmaceutics-13-01797-t005:** Processing time during dry mixing stages of amlodipine besylate layer at different locations (L) in batch 1.

Stage	Time (min)	Amlodipine Besylate Content (%)							
		L1	L2	L3	L4	L5	L6	Mean	CV (%)
Dry mixing	3	102.04	101.79	104.43	99.78	101.54	102.89	102.08	1.51
	5	101.06	101.20	100.81	100.09	101.94	100.38	100.91	0.71
	7	103.75	102.58	100.26	101.37	102.71	102.65	102.22	1.19
Dry mixing Completion	3	107.74	103.85	102.54	106.32	104.88	100.78	104.35	2.42
	5	102.61	104.00	104.18	106.01	104.46	106.07	104.56	1.20
	7	104.71	105.20	106.02	101.81	100.76	103.06	103.59	1.99

**Table 6 pharmaceutics-13-01797-t006:** Optimal process of amlodipine besylate layer in three batches.

Stage	Batch	Amlodipine Besylate Content (%)							
		L1	L2	L3	L4	L5	L6	Mean	CV (%)
Dry mixing	1	101.06	101.20	100.81	100.09	101.94	100.38	100.86	1.38
	2	101.46	101.61	100.43	100.95	101.58	101.83	101.31	0.59
	3	102.24	101.07	101.91	101.20	100.67	102.49	101.60	1.09
Dry mixing Completion	1	102.61	104.00	104.18	106.01	104.46	106.07	104.56	1.20
	2	102.71	100.2	104.02	101.81	100.76	102.06	101.93	1.25
	3	99.17	100.6	98.46	102.37	101.03	101.31	100.49	1.31

**Table 7 pharmaceutics-13-01797-t007:** Evaluation parameters of granules.

Active Ingredient	Batch	Moisture Content (%)	Bulk Density (g/mL)	Tapped Density (g/mL)	CI (%)	Hausner’s Ratio	Angle of Repose
Metoprolol succinate	1	2.590 ± 0.260	0.461 ± 0.005	0.500 ± 0.009	7.801 ± 0.055	1.085 ± 0.041	23.38 ± 0.640
	2	2.680 ± 0.440	0.460 ± 0.002	0.499 ± 0.189	7.815 ± 0.042	1.084 ± 0.027	23.39 ± 0.370
	3	2.610 ± 0.260	0.459 ± 0.003	0.499 ± 0.094	8.016 ± 0.078	1.087 ± 0.049	23.35 ± 0.510
Amlodipine besylate	1	-	0.754 ± 0.012	0.833 ± 0.150	9.483 ± 0.026	1.105 ± 0.103	28.97 ± 0.650
	2	-	0.754 ± 0.023	0.833 ± 0.113	9.484 ± 0.120	1.105 ± 0.055	28.97 ± 0.470
	3	-	0.755 ± 0.091	0.834 ± 0.057	9.472 ± 0.108	1.105 ± 0.078	28.99 ± 0.190

All values are the mean ± standard deviation (SD) for *n* = 3.

**Table 8 pharmaceutics-13-01797-t008:** Results of the investigated Shewhart R chart for the tablet weight of three batches.

Rule	Request (Not Allowed)	Results		
		Batch 1	Batch 2	Batch 3
Control limit +3σ	1 point beyond the control limit +3σ	Reach	Reach	Reach
High value zone	6, 7, 8, 9 consecutive points above $\bar{R}$	Reach	Reach	Reach
Trends up	6, 7, 8, 9 consecutive points trending up	Reach	Reach	Reach
Control limit +3σ	2 out of 3 points in the zone +A 3 out of 7 points in the zone +A 4 out of 10 points in the zone +A 4 out of 5 points in the zone +B	Reach	Reach	Reach

**Table 9 pharmaceutics-13-01797-t009:** Results of the investigated Shewhart $\bar{X}$ chart for the tablet weight of three batches.

Rule	Request (Not Allowed)	Results		
		Batch 1	Batch 2	Batch 3
Control limit +3σ	1 point beyond the control limit +3σ	Reach	Reach	Reach
Control limit −3σ	1 point beyond the control limit −3σ	Reach	Reach	Reach
High value zone	6, 7, 8, 9 consecutive points above $\bar{X}\;$	Reach	Reach	Reach
Low value zone	6, 7, 8, 9 consecutive points below $\bar{X}\;$	Reach	Reach	Reach
Trends up	6, 7, 8, 9 consecutive points trending up	Reach	Reach	Reach
Trends down	6, 7, 8, 9 consecutive points trending down	Reach	Reach	Reach
Control limit +3σ	2 out of 3 points in the zone +A 3 out of 7 points in the zone +A 4 out of 10 points in the zone +A 4 out of 5 points in the zone +A	Reach	Reach	Reach
Control limit −3σ	2 out of 3 points in the zone −A 3 out of 7 points in the zone −A 4 out of 10 points in the zone −A 4 out of 5 points in the zone −B	Reach	Reach	Reach

**Table 10 pharmaceutics-13-01797-t010:** Evaluation of film coated bi-layer tablet of three batches.

Batch	Appearance	Weight Variation (mg) (*n* = 20)	Content Uniformity (mg) (*n* = 10)	Hardness (kp) (*n* = 10)	Friability (%) (*n* = 17)	Drug Content (%)	
						Metoprolol Succinate (*n* = 20)	Amlodipine Besylate (*n* = 20)
1	White coloured, oval shaped, biconvex tablets with smooth surface	408.82 ± 0.57	4.72 ± 3.56	10.48 ± 0.57	0.14 ± 0.03	94.52 ± 0.86	96.50 ± 1.12
2		411.72 ± 0.84	4.87 ± 2.43	10.44 ± 0.15	0.13 ± 0.06	93.96 ± 0.61	95.12 ± 1.10
3		411.32 ± 0.95	4.88 ± 2.42	10.36 ± 0.68	0.14 ± 0.25	93.56 ± 1.41	96.76 ± 0.95

All values are mean ± standard deviation (SD).

**Table 11 pharmaceutics-13-01797-t011:** Dissolution test of film-coated bi-layer tablet of three batches.

Batch	Active Ingredient	Cumulative Drug Release Percentage (%)				
		30 min	1 h	4 h	8 h	20 h
1	Metoprolol succinate	-	13.99 ± 2.99	35.22 ± 2.83	52.28 ± 3.03	87.25 ± 2.16
	Amlodipine besylate	98.66 ± 2.61	-	-	-	-
2	Metoprolol succinate	-	14.01 ± 1.15	35.93 ± 0.94	52.12 ± 1.71	86.21 ± 0.96
	Amlodipine besylate	96.23 ± 1.03	-	-	-	-
3	Metoprolol succinate	-	12.69 ± 1.68	34.07 ± 1.10	51.50 ± 1.08	86.55 ± 0.81
	Amlodipine besylate	94.39 ± 1.98	-	-	-	-

All values are the mean ± standard deviation (SD) for n = 6.

## Data Availability

Not applicable.

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
