# Peer review of "Development of a Scalable Process of Film-Coated bi-Layer Tablet Containing Sustained-Release Metoprolol Succinate and Immediate-Release Amlodipine Besylate"

_pharmaceutics, 2021, doi:10.3390/pharmaceutics13111797_

Round 1
Reviewer 1 Report
The manuscript by Tuyen et. al reported the scale-up process of the bi-layer metoprolol and amlodipine tablet. The authors performed dry mixing, drying, and dry mixing completion phase optimization. The physical parameters were tested and in vitro dissolution was observed up to 20 h. The incorporation of Shewhart R chart and chart in scale-up process is interesting, However, the reviewer could not find the novelty in the scope of study. The authors could have opted for QbD design for optimization or have chosen simulation of dry mixing or drying method. The inclusion of mathematical models could have been an advantage for the sustained-release metoprolol. The authors cited reference (6), which already conducted the similar bi-layer tablet of metoprolol and amlodipine. The scale-up of any lab scale formulation to pilot batch (1/8th of lab scale or 10,000 units) or commercial batch (can be any larger batch size) are covered in the prospective concurrent validation (n=3 batches), as per the GMP guidelines and the respective country’s regulatory requirements. Hence, it may not contribute as break through concept in the field of scientific research and academic society. Moreover, the discussion was not supported with adequate experimental data and citations. The authors claimed that an increased hardness could reduce the friability based on a reference (29) but the cited reference is not about a study of the bilayer tablet. The authors are requested to justify the batch size of granules with the volume of high-speed mixer granulator as the batch size of 4 kg seems approximately 80% fill volume for the 5 kg high-speed mixer. In page 3, Table 1, the authors mentioned Opadry II 85F19250 Clear as formulation ingredient. The reviewer could not understand why the authors used film coating agent in the formulation, as it was not discussed elsewhere. The authors pre-compressed metoprolol granules at 3 KP and compressed with amlodipine granules. The authors are suggested to include additional discussion about compression force necessary for amlodipine layer, and how it affected the bonding of the two layers. The authors are suggested to include the control sample (SelomaxTM, SitelolAM) for comparing the in vitro dissolution of the present study. There are typos and grammatical errors. In conclusion, the reviewer does not recommend the manuscript for the publication in present form.
Please, go through the specific comments below:
- In line 41, the authors mentioned ‘…..to plant life.’ It is not clear how the high blood pressure affected the plant life. It is not related to the manuscript title. The authors are suggested to revise the portion.
- In line 44 and 45, the authors mentioned ‘….European Association of Hypertension (ESH) and the European Heart Association (ESC)’. The reviewer suggests as ‘European Society of Cardiology and European Society of Hypertension (ESC/ESH)’.
- In line 69, the authors are suggested to recheck ‘SitelolAM’. The superscription of ‘AM’ may not be required as the reviewer feels ‘AM’ stands for amlodipine. It is different from the case ‘SelomaxTM’ where TM represents trademark.
- In page 3, Table 1, line 109, the authors mentioned Opadry II 85F19250 Clear as formulation ingredient. It is confusing why the authors opted to use the film coating agent in the bi-layer tablet formulation, as it was not discussed elsewhere. The use of xanthum gum is also not clear. The PVP K30 as binder and HPMC K100M as sustaining agent used in metoprolol layer while amlodipine layer is directly compressed.
- In line 117, the sentence ‘……magnesium stearate were accurately weighed and passed through a sieve (#50-mesh) to ensure a disaggregated state prior to mixing’ is confusing. Have the authors granulated magnesium stearate along with the other excipients. In that case, it is incorrect to process in such a way.
- The excipient ‘Glucidex’ is not mentioned elsewhere beside in page 3, Table 1.
- In line 124, high-speed mixer NT-5 capacity mentioned was 5 kg and the weight of metoprolol succinate and other excipients in Table 1 was nearly 4.06 kg, approximately 80% capacity of the high-speed mixer. The optimum granulation volume is considered 30% to 60%. The authors are requested to justify in this aspect.
- The reference 13 and reference 14 were not cited whereas reference 15 was not cited in order in the manuscript.
- Reference 16 and reference 17 could not be accessed. The cited references are not available for the readers. Therefore, the HPLC validation parameters could not be observed. The authors could briefly include the parameters, a range and limits.
- The authors are suggested to provide the coefficient of variance (CV) equation after line 147.
- Line 155, the authors are requested to explain the word ‘aqueous acid’ (pH=4). It is hard for the readers to understand which aqueous acid.
- The authors are suggested to label the six different inserts in the Figure 1 as (A), (B), (C), (D), (E), and (F), which would make the readers to understand the discussion easily.
- In line 181 to 194, rule 1 to 6, the authors are suggested to explain in a simpler way. Maybe the graphical representation could be easier to understand.
- In line 201, the authors mentioned ‘digital precision balance (Mettler Toledo PL202-S, 201 Switzerland, readability of 0.01 g)’ for tablet weight measurements. In reviewer’s opinion, if the authors were measuring individual tablet weights (250 mg, 400 mg), readability of 0.001 g could have been better. In case of average weight of 20 tablets (8 g), the balance of 0.01 g is satisfactory.
- In line 378, a sentence mentioned ‘The study has reported that there will be a decrease in the percentage of friability with an increase in the tablet hardness’. The cited reference (29) for the discussion, studied a conventional tablet whereas the authors studied a bi-layered tablet. In reviewer’s opinion, the conventional single layer tablet and bi-layered tablet are different. So, the statement could not be correlated.
Author Response
Dear Reviewer,
We would like to thank the reviewers for your critical questions and comments, which indeed help us tremendously in improving our manuscript.
The questions of the reviewer, our responses, and English revision are addressed in the attached “Reply to the Reviewer” and incorporated in the revised manuscript.
Sincerely Yours,
Nguyen Thi Linh Tuyen

Reviewer 2 Report
The paper presented by the authors is a very thorough study important subject area. It would be highly relevant to the readership of pharmaceutics and therefore the paper should be published. But before publication the following points need to be addressed.
One) the abstract is well written and summarises the work expertly. The only point here would be to define the meaning of the error, is this a percent CV, standard deviation et cetera. Please define how the plus and -4% was determined.
Two) the introduction is very good and nicely summarises the relevant literature. To improve please add a clear aim in of the introduction.
Three). In the material section please provide the purities where available and the batch numbers for the chemical consumables used.
Four). The information in table 1 clearly outlines very accurate measurements for weight, it would be good in the table legend or beneath the table to indicate the likely fair experimental error of the weight measurements.
Five). Line 120 please give the temperature range for the ambient temperature described.
Six). Line 126 please describe a little bit more information concerning the operation of the fluidised bed dryer. What were the temperatures, flow rates and gases used. Was it hot air or hot nitrogen for example?
Seven). A HPLC assay was used for content uniformity, drug content and dissolution. What evidence is there that is able to detect degradation products.
Eight). In table 7 the moisture content is described please give a commentary on how this was determined in the method section.
Nine). Would it be possible to add a table showing pictures of the different tablets produced as there are observational results given in table 10. And it would be good to see how these correlate to the images of the tablets.
Ten). The paper is really good, and an interesting read, but please in the conclusion add an extra sentence highlighting whether the aim has been fulfilled and the novelty of the study.
Author Response

(The authors gave the same response as above.)

Reviewer 3 Report
This manuscript provides the methods to the scale-up process of bi-layer tablet containing sustained-release metoprolol succinate and immediate-release amlodipine besylate. Overall, the work is interesting for the readers of Pharmaceutics. However, two issues should be addressed before it is considered for publication:
Line 217: Section: Dissolution studies. Authors should better describe how were the dissolution media changed in the continuous dissolution test.
Lines 330-33, 364, 366, 376 etc: Kilopond units are usually written in small letters kp
Author Response

(The authors gave the same response as above.)

Reviewer 4 Report
The manuscript "Scale-up process of bilayer tablet containing ..." describes rather a manufacturing report than a scientific study. The only systematic variation assessable to the reader is the blending/drying time. Accordingly, the titel is completely misleading. This study does is not of interest for the scientific community, addressed by MDPI pharmaceutics, as the results seem not transferable to other cases and are not thoroughly discussed. Additionally, the language of the manuscript does not add to the understanding of the topic and is often misleading.
Points that support my decision:
- Not scale-up is presented, a smaller scale (and data of this) is missing
- The aim and the study are not scientific, rather political or economical
- Unsuitable, non-scientific units (e.g. KP) are used
- Lack of novelty
- Lack of transferability, presenting only a singular solution
Accordingly, to my opinion, only a highly applied, industrial journal can reach a community that may be interested in the results presented.
Author Response

(The authors gave the same response as above.)

Round 2
Reviewer 1 Report
Since the authors answered the questions in reasonable ways, the reviewer would like to recommend the manuscript as acceptable for the journal.
Author Response
Thank you very much for your professional comments and questions to help us substantially improve our work, and sincerely thank for your “accept” recommendation.
Reviewer 4 Report
Dear authors, thank you for further developing your manuscript. However, I do not see my major concern settled that the novelty and transferability of the content justiefies a publication. It is a now well written and well discussed production report, but the contribution to the community is very minor in my opinion.
A major example of unprecise preparation is additionally that bilayer tablets (9x12 mm, oval) should be compressed with a force of 100-120 N. This is impossible as it would mean 0.1 kN and by that approx. 1 MPa. The formulation would be ground-breaking if that was right.
Accordingly, I personally keep up my judgement to reject the manuscript.
Author Response
Thank you very much for recognizing our efforts of improving the manuscript followed your professional comments!. We do want to have better contribution to the community, however, the limitations of our budget and laboratory conditions constrain us to do so. We will keep in minds your comments to improve our future works.
The first author terribly sorry for the serious error on writing the unit of the compression force. The first author has double checked the unit, and she found that she was wrong. Instead of writing 10 – 12 kN, she noted as 10 – 12 kp. Therefore, the first version of the manuscript reported the compression force of 10 – 12 kp. Then, the reviewer #4 noted “the unit was not suitable”, we wrote both the force units in kp and N as “…pressing force 98.1 – 117.7 N (or 10 - 12 kp)” in the revised manuscript. Briefly, we confirm that the compression force was 10 – 12 kN, and we corrected the error in the revised manuscript accordingly.
